# Effectiveness of a Therapeutic Educational Oral Health Program for Persons with Schizophrenia: A Cluster Randomized Controlled Trial and Qualitative Approach

**DOI:** 10.3390/healthcare11131947

**Published:** 2023-07-05

**Authors:** Frederic Denis, Corinne Rat, Lucie Cros, Valerie Bertaud, Wissam El-Hage, Lysiane Jonval, Agnès Soudry-Faure

**Affiliations:** 1Faculty of Dentistry, Tours University, 37000 Tours, France; 2EA 75-05 Education, Ethics, Health, Faculty of Medicine, François-Rabelais University, 37000 Tours, France; 3Clinical Research Unit, La Chartreuse Psychiatric Center, 21033 Dijon, France; corinne.rat@chlcdijon.fr; 4Instance Régionale d’Education et Promotion de la Santé, 76100 Rouen, France; l.cros@ireps-bfc.org; 5Health Big Data, LTSI-INSERM U 1099, University of Rennes 1, 35043 Rennes, France; valerie.bertaud@univ-rennes1.fr; 6Rennes University Hospital, Guillaume Regnier Hospital, 35700 Rennes, France; 7CIC 1415, U 1253 iBrain, Institut National de la Santé et de la Recherche Médicale (INSERM), Centre Hospitalier Régional Universitaire (CHRU), 37000 Tours, France; wissam.elhage@univ-tours.fr; 8USMR-Réseau d’Aide Méthodologiste, University Hospital of Dijon, CEDEX, 21079 Dijon, France; lysiane.jonval@chu-dijon.fr (L.J.); agnes.soudry@chu-dijon.fr (A.S.-F.)

**Keywords:** schizophrenia, oral health, therapeutic educational program, dental education

## Abstract

Background: The oral health of people with schizophrenia (PWS) is very poor, suggesting a need for oral health promotion programmes with a high level of evidence. The aim of the EBENE study (Clinicaltrials.gov: NCT02512367) was to develop and evaluate the effectiveness of a multidisciplinary therapeutic educational programme in oral health (TEPOH) for PWS. Methods: A multicentre cluster randomised controlled trial, with outpatient psychiatry centres as the unit of randomisation, was designed to compare the effectiveness of TEPOH (intervention group) versus standard care (control group). The trial was conducted in 26 outpatient psychiatry centres in France (14 in the intervention group, 12 in the control group). Eligible patients with a diagnosis of schizophrenia were enroled between 2016 and 2020 and followed for 6 months. The TEPOH group received a multicomponent intervention (comprising an introductory session, three educational sessions, and a debriefing session). The primary endpoint was the evaluation of periodontal disease as a community periodontal index (CPI) score ≥ 3 at Month 6. The trial was completed using a qualitative approach based on semi-structured interviews with caregivers conducted between July 2018 and December 2019. The trial was stopped early due to difficulties in recruiting patients. Results: Overall, 81 patients (of 250 planned) were included, and 54 patients completed the trial: 40 in the TEPOH group and 14 in the control group. At baseline, the percentage of CPI ≥ 3 was 42.5% in the TEPOH group and 9.1% in the control group. At Month 6, the percentage of CPI ≥ 3 was 20% in the TEPOH group and 14.3% in the control group. The qualitative evaluation underlined that the professionals emphasised the “seriousness” and “assiduity” of the patients’ participation in this programme and that the TEPOH reinforced carers’ investment in oral hygiene. It also highlighted structural factors (lack of resources for professionals, lack of teeth in PWS, COVID-19 pandemic) that may have exacerbated the difficulties with enrolment and follow-up. Conclusions: The effectiveness of this TEPOH, developed for PWS as part of the EBENE study, has not been demonstrated. Certain aspects of the programme’s content and implementation need to be reconsidered. In particular, an adapted subjective measurement scale should be developed.

## 1. Introduction

People with schizophrenia (PWS) are often affected by poor oral health [1]. Dental caries and periodontal disease are generally more prevalent in these populations than in the general population [2,3]. One reason for this is the negative symptoms of the disease. Among these symptoms, apathy and emotional blunting are responsible for social withdrawal, impaired social performance, and/or self-care [4,5]. More generally, the symptoms of schizophrenia lead to impaired thought progression, contextual analysis errors, and logical errors. In this context, PWS do not sufficiently recognise their health needs and delay seeking advice or treatment [6]. Daily oral hygiene or regular visits to the dentist often take a back seat to other lifestyle needs. In addition, the side effects of treatment combined with poor dietary and lifestyle habits (high sugar diet, use of psychoactive substances such as tobacco, poor oral hygiene) make the situation worse [7,8]. More generally, 19–57% of people with severe mental illness have numerous co-morbidities (cardiovascular, gastrointestinal, respiratory, neoplastic, infectious, endocrine, and oral disorders). It is estimated that half of these co-morbid conditions are undiagnosed [9,10].

As a severe and persistent mental disorder, schizophrenia affects about 1% of the world’s population and 600,000 people in France [11]. There is, therefore, a public health issue in developing oral health promotion and prevention actions adapted to these populations. The difficulty in building these programmes is linked to the discrepancies observed between the expectations or perceptions of health needs expressed by the people themselves and by health professionals or other social workers [12,13]. Between the recognition of a health need and the actual use of care, between the knowledge of risk and the adoption of health-promoting behaviours, there is a whole trajectory that takes place for each individual and that needs to be put into perspective in order to better adapt responses to needs [14]. In the case of PWS, these trajectories are difficult to decipher because of the complexity of the situations induced by the mental disorder.

We hypothesised that a multidisciplinary therapeutic education programme taking into account the views of dentists, psychiatrists, nurses, doctors, psychologists, PWS, carers, and therapeutic education specialists, would be likely to have a more positive impact on the oral health of PWS. Co-construction with all stakeholders is a guarantee that the approach will be adopted in the development of day-to-day care practices. [15].

In this context, the EBENE study was funded by the French Ministry of Health (Direction Générale de l’Offre de Soins) and aimed to build and evaluate the effectiveness of an oral health education programme adapted to PWS. The study protocol has already been published [16], and the EBENE study comprises three stages. The first stage of the study was to construct the program with the participation of PWS as research partners and not as research subjects and all respondents (dentists, psychiatrists, nurses, doctors, psychologists, caregivers, and specialists in therapeutic education). The development of the programme content was qualitatively studied in the form of a focus group (FG) [17].

The second stage was to test the feasibility of this programme in a cluster randomised controlled trial [16].

The third stage of the EBENE study was to evaluate the effectiveness of this therapeutic educational programme in oral health (TEPOH) in the context of a randomized controlled cluster study using a population with schizophrenia sample recruited from patients in psychiatric hospitals in France and to evaluate changes in caregivers’ perceptions and behaviours regarding oral health in PWS. The aim of this study was to present the results of the third stage of the EBENE study. A cluster randomized controlled trial was completed using a qualitative approach. 

## 2. Materials and Methods

### 2.1. Research Study Design

This is a cluster randomised, controlled, multicentre, open-label intervention trial with a 6-month patient follow-up period. This intervention trial assessed the effectiveness of TEPOH in PWS followed in 26 outpatient psychiatry centres in France Clinicaltrials.gov: NCT02512367). A cluster design was chosen as the intervention was implemented in a team setting, and to minimize contamination between the intervention and control groups (TEPOH vs. standard care); clusters were defined as outpatient psychiatry centres. The trial was completed with a qualitative approach based on semi-directive interviews of caregivers to complement the results of the main study. These interviews aimed to explore the view that caregivers had a posteriori on the implementation of the EBENE program within their structure.

The trial design has been reported previously [16]. There were major changes to the study design published previously after the results of the feasibility study: (1) change of study setting in order to be able to assess stabilized schizophrenic patients receiving outpatient care, (2) extension of the inclusion criteria to protected adult patients (target population), (3) revision of the program period and follow-up visit (education program of 2 months instead of 6 months, intermediate evaluation at Month 3 and final evaluation at Month 6 instead of Months 6 and 12, respectively), (4) deletion of the stratum of randomization, (5) increase in the number of patients to be included, (6) replacement of the WHO-QOL questionnaire by S-QOL, (7) replacement of the PANSS questionnaire by BECK. These changes occurred before the inclusion of patients.

Overall, patients were enroled between 2 November 2016 and 13 January 2020, with the last patient’s final visit on 5 October 2020. The trial was stopped early due to difficulties in recruiting patients: i.e., selecting patients in compliance with inclusion and exclusion criteria.

### 2.2. Intervention

Our intervention was a therapeutic educational program in oral health designed as follows: (1) An introductory session with staff, caregivers, and patients, (2) three educational sessions for groups of 5–6 patients, each lasting 90 min, spaced 2 weeks apart, for 2 months, and (3) a debriefing session with patients at the end of the program. The sessions were co-led by a caregiver of the study centre and a professional of IREPS BFC (Instance Régionale d’Education et de Promotion de la Santé de Bourgogne Franche-Comté). The educational intervention is a process of reinforcing the patient’s ability to manage their oral hygiene. Educational intervention is an integral part of therapeutic management. It is complementary to and inseparable from treatment and care. The effectiveness of the educational intervention is evaluated using various criteria described in Section 2.5.

The construction of the programme has been published previously [18,19]. Three distinct educational themes were identified. The first theme was to mobilise motivational approaches by improving self-esteem and well-being and was called “Yes we can.” The second theme aimed to demystify dental surgery and was called “Even more afraid.” The third theme aimed to improve oral health through a cross-cutting approach to quality of life (smoking cessation, diabetes control, managing a healthy diet, etc.) and was called “Taking care of myself.”

### 2.3. Recruitment of Centres and Participants

All psychiatry centres and their annexed facilities providing care for PWS were eligible for inclusion (Medical-Psychological Centre (CMP), Part-Time Therapeutic Reception Centre (CATTP), or Day psychiatric Hospital (Hospital de Jour, HJ). Two authors (FD and CR) contacted eligible psychiatry centres between June 2016 and November 2016 on the basis of geographical criteria and their ability to be integrated into a research process. The centres that agreed to participate were randomized into the intervention or control group before participating in the trial (TEPOH versus standard).

In each centre, patient eligibility was determined according to the inclusion and exclusion criteria described in Table 1, i.e., persons with a diagnosis of schizophrenia as defined in the Diagnostic and Statistical Manual of Mental Disorders-Fifth edition (DSM-5) [20] and stabilized from a psychiatric viewpoint. The selection was made by the psychiatrist at the participating centres. Prior to inclusion, each patient signed a written consent form.

### 2.4. Randomization

Before participating, psychiatry centres were randomized with a 1:1 ratio into the intervention or control group by a person from the coordination centre that was not involved in the study, using the secure, internet-based Tenalea™ software (Formsvision BV, Abcoude, The Netherlands, version 3, https://prod.tenalea.net/gso/dm/). The treatment algorithm was generated by the statistician from the coordination centre before the start of the trial (USMR, Dijon, France). The allocation was based on a minimization approach considering the region.

Due to the nature of the study intervention, it was impossible to blind patients or assessors of oral health measurements to the assignment of the intervention. Nevertheless, randomization was revealed after the recruitment of the centres to ensure the concealment of allocation.

### 2.5. Outcomes

The primary endpoint was the evaluation of periodontal disease using a rate of community periodontal index (CPI) ≥3 at Month 6 [18]. The CPI index quantifies the degree to which the periodontium is affected by periodontal disease. It is the benchmark indicator of periodontal health and is closely linked to patients’ level of hygiene [21].

The secondary endpoint was the rate of CPI ≥ 3 at Month 3, the mean change in the decayed, missing, and filled teeth (DMFT) score from baseline to Months 3 and 6, and the mean change in the hygiene index (OHI-S) score from baseline to Months 3 and 6 [22]. The DMFT index gives the sum of an individual’s decayed, missing, and filled permanent teeth and provides an indicator of both current and past caries experience [23]. The OHI-S indices help dentists to determine a patient’s level of oral hygiene by scoring debris and calculus accumulation in the mouth. For all of the dental examinations above, the dental specialists have been calibrated against the chief investigator through repeated examinations of a separate pilot sample using similar indices, followed by meetings to discuss discrepancies and standardize procedures. Kappa scores of 0.9 for an inter-rater agreement were achieved.

Other secondary evaluation endpoints included the mean change in the quality of life from baseline to Months 3 and 6 using the French version of the Global Oral Health Assessment Index (GOHAI) self-report scale [24] and the Schizophrenia Quality of Life Scale, S-QoL [25] and the mean change in depressive symptoms from baseline to Months 3 and 6 according to the Beck Depression Inventory [26]. All scales used are shown in Table 2.

Depending on the protocol, the number of PWS to be included with a type-I error of 5% and a type-II error of 20% was set to 250 patients to highlight a statistically significant difference between the two groups [16].

### 2.6. Analysis

EBENE was a superiority trial. The sample size was based on the primary outcome. Considering that the rate of CPI ≥ 3 at Month 6 would be 20 percentage points lower in the TEPOH group than in the standard group (20% vs. 40%) and assuming a cluster variability (CV = 0.46) and intra-class correlation coefficient (CCI = 0.01) and that 10% of the patients would not be able to be evaluated, we estimated that we would need to enrol 250 patients (125 per group) to provide 80% power with a two-sided alpha level of 0.05 [16].

The primary analysis was performed according to the intent-to-treat principle and included all randomized patients. The baseline patient characteristics and outcomes were described by mean ± standard deviation for quantitative variables and by frequency (%) for qualitative variables. We calculated the rate of CPI ≥ 3 at Month 6 (primary outcome). As the trial was stopped early, only descriptive analyses were realized at baseline, 3 and 6 months. All analyses were performed using SAS version 9.4 (SAS Institute INC.).

To complete the quantitative analysis, a health sociologist conducted exploratory interviews with health managers and nurses. The objective of these interviews was to highlight the grey areas of this study concerning the strengths and weaknesses of the programme from the point of view of these professionals. Only a health professional coordinator and nurses were questioned. The doctors, after agreeing to the participation of the patients in this study, were not involved in the implementation of the programme. These interviews were conducted using qualitative research methodologies [27].

### 2.7. Ethics

EBENE was funded by the “Programme Hospitalier Recherche Clinique” research program, and Dijon University Hospital was the sponsor. The study was approved by the institutional review board (French Ethics Committee for the Protection of Persons (CPP) III of Eastern France (Approval reference: 2015-A00407-42) and the French data protection agency (CNIL), complied with the ethical standards defined by the Declaration of Helsinki and Good Clinical Practice, and was registered on Clinicaltrials.gov: NCT02512367.

## 3. Results

### 3.1. Results of the Cluster-Randomised Controlled Trial

In total, 26 psychiatric centres were randomized into TEPOH or standard groups. Among the 26 centres, 12 did not recruit patients. In the TEPOH group, 51 patients were recruited: 47 patients completed the baseline visit, 31 (60.7%) patients completed the Month 3 visit, and 40 (78.4%) patients completed the Month 6 visit (Figure 1). In the standard care group, 30 patients were recruited: 22 patients completed the baseline visit, 12 (40.0%) patients completed the Month 3 visit, and 14 (46.7%) patients completed the Month 6 visit. The distribution of this population is shown in the diagram below (Figure 1).

### 3.2. Description of Study Population for the Cluster Randomized Controlled Trial

The table below shows the characteristics of the study population (Table 3). At baseline, the overall age was 44.4 ± 11.6 years, and 36.2% of participants were female. Lifestyle was similar between groups for the consumption of soft drinks, alcohol, tobacco use, and snacking between meals. Participants in the intervention group lived more frequently at home than participants in the control group.

The two groups had similar baseline oral and dental health status scores (CPI, CPITN, DMFT, OHIS), but differences were observed between the groups when considering the sub-scores for DMFT (decayed teeth in the intervention group and missing teeth in the control group) or the measurement in category for CPI (rate of CPI ≥ 3 higher in the intervention group than in the control group).

### 3.3. Description of the Assessment Criteria

The percentage of CPI ≥ 3 at Month 6 was 20% in the TEPOH group and 14.3% in the standard group.

There were no significant differences between groups for secondary endpoints (Table 4).

### 3.4. Results of the Qualitative Study

The survey took place between July 2018 and December 2019. Four exploratory interviews were conducted with a health executive and with three nurses. It was possible to distinguish elements relating to the organisation and coordination of the programme, which are important for the study’s feasibility. This approach also made it possible to highlight the way in which patients were mobilised to participate in the programme and to assess the impact of this programme in a subjective way.

#### 3.4.1. The Organization and Coordination of the Programme

The programme implementation and patient recruitment phase were particularly demanding in terms of inclusion criteria. Professionals reported difficulties in recruiting patients who met the criteria for participation in the study.


*We included people but they were at the limit of the study. (…) I know that we had difficulty finding patients who met the criteria. (…) There were those who met the criteria for the number of teeth but some were too delirious to answer the questionnaires.*
(A nurse)

The professional who was most involved in the study was the health care manager who coordinated the patients, doctor, and health care team.


*Our manager really carried the organisation of this study at arm’s length…it took a lot of work to contact the patients, to make the panoramic X-rays, to organise the appointments.*
(A nurse)

We observed that the manager was responsible for the internal coordination of the TEPOH, as well as for managing the logistical aspects: planning the sessions, booking the rooms, checking the availability of at least one staff member, the availability of patients, etc. Despite this set of criteria being combined, the speeches seemed to be marked by an idea of ease in implementing the TEPOH.


*There were really no difficulties in setting up this Ebene Protocol. All the documents we had were very clear, and the patients agreed… There was really no difficulty at all.*
(A nurse)

The TEPOH sessions were conducted every fortnight. This pre-established rhythm in terms of the spacing of sessions was perceived favourably.


*Yes, it was good because with our patients it should not be too close or too far away because after a while, they forget…*
(A health care manager)

#### 3.4.2. The Way in Which the Patients Mobilized for the Program

The professionals testified to the “seriousness” and “assiduity” with which the patients participated in this programme. Even if the carers occasionally had to remind the patients of certain appointments.


*They were actually happy. Some of them were obliged to come in addition to what they usually did in the day hospital. We didn’t really need to remind them…it showed their motivation, because we ourselves, for certain types of care, when they are not motivated, they find excuses not to come… Whereas in your study, I think there were never any absentees… This is an indicator that shows that they were happy with this care.*
(A health care manager)


*Everyone really took their appointments with great seriousness. With this pride in brushing their teeth well, in maintaining the level of hygiene.*
(A Nurse)

#### 3.4.3. The Programme Effects

This TEPOH was based on the creation of favourable environments, which implies taking into account living conditions, for example, access to oral hygiene equipment. In this case, the program would not have significantly modified the patient reception environment. Professionals point out, for example, that oral hygiene equipment was already available to patients.


*We give toothbrushes …so there are already things that exist… In most bathrooms they have posters on how to brush their teeth.*
(A Nurse)

Among the possible effects of the program on the patients, one hoped for a reduction in the consumption of sweets and tobacco or more frequent and regular brushing of the teeth. It appears that the professionals interviewed did not observe much effect of the program on the daily oral hygiene of the patients, even if some patients made greater efforts during the oral health promotion sessions.


*I didn’t see any effect on the patients, and when I talked to the patients who are part of the study, there was no change in their tooth brushing habits… we tried to mobilise them… they say ‘yes we know’ but they didn’t do it.*
(A Nurse)

This being the case, it seems that the logic of change can still be activated, sometimes in a more occasional manner, and was often subject to more individualised follow-up with the patient.


*For some it was easier to go to the dentist after the TEPOH. So that’s already a big positive… It took away the fear of the dentist…*
(A health care manager)

The programme was reported to facilitate the provision of dental care and follow-up significantly. In this sense, the EBENE TEPOH seems to foster the development of individual skills in patients who participated in the intervention sessions.


*It was interesting because they were proud afterwards to have maintained their level of good dental health.*
(A Nurse)

While health professionals clearly did not wait for the EBENE programme to address oral health, the programme reinforced carers’ investment in oral hygiene, making them more supportive of existing practices.


*it made it possible, after EBENE, to propose to a patient at least once a week to brush his teeth.…*
(A Nurse)

The programme also works on the caregivers’ perceptions of oral hygiene. This can have a positive effect on their ability to make suggestions. For example, they could provide individual support for teeth brushing.


*The idea is that what we bring to them in psychoeducation in our workshops has an impact at home.*
(A Nurse)

## 4. Discussion

The objective of this study was to evaluate the effectiveness of a TEPOH with a study design that would provide a high level of evidence on the relevance of a programme co-constructed with patients suffering from schizophrenia and based on an empowerment process. Although our research study did not show TEPOH to be superior to standard care regarding CPI ≥ 3 and dental evaluation scores, it is interesting to look back at what we have learned and the different limiting factors of this study.

First, enrolment and follow-up were difficult, and the number of participants was affected significantly. During enrolment, the patients assessed for eligibility did not meet the inclusion criteria because of a lack of teeth in this population. At follow-up, it is probable that the duration of the study (6 months) was incompatible with the symptoms of schizophrenia. The many symptoms of schizophrenia, such as ambivalence, apragmatism, executive function disorders, hallucinations, and delusions, which are combined to varying degrees of intensity with negative disorders such as self-absorption, make it difficult to engage in long-term actions on a daily basis [28,29]. In addition to the necessary availability of PWS, which is often incompatible with their psychological disorder, we can assume that there is a relationship between the participation of the patients and the investment of the healthcare team to accompany them in this research protocol in view of the difficulties they encounter in accessing the health care system in general [30].

This qualitative study pointed out that the lack of projection in the reproducibility of the programme was not directly linked to the content of the sessions or a lack of interest but rather to structural factors. For example, the lack of resources could explain a weakness in the approval of the facilitation techniques and intervention tools used in the programme. In this respect, none of the carers interviewed indicated that they had at any time reused these techniques or tools or been inspired by them to create a workshop. At the same time, no one mentioned any lack of training or at least a lack of training that would penalise the replicability of the programme. However, the cross-disciplinary approach to oral health, i.e., the approach based on common determinants such as tobacco and/or diet, as recommended by the WHO, is not well developed in the general population and even less so in psychiatry, where care focuses on stabilising the symptoms of mental illness [31]. In the qualitative approach, the professionals emphasised the “seriousness” and “assiduity” of the patients’ participation in this programme. The TEPOH reinforced carers’ investment in oral hygiene, making them more supportive of existing practices, and the EBENE TEPOH seems to foster the development of individual skills in patients who participated in the intervention sessions. The co-construction of this programme has given it meaning and reinforced the value of this approach, including with PWS. This co-construction approach, which integrates the principles of health promotion [32] and social cognitive theories [33], was used to develop TEOHP [16].

The TEPOH appears to have had a beneficial effect on carers’ perception of oral hygiene. They emphasised that, thanks to the programme, they were able to provide individual support in brushing their teeth, provided there were enough of them to perform this task. Generally speaking, the field of psychiatry suffers from a profound lack of human and financial resources in France, despite the fact that the demand for care is constantly increasing [30].

More generally, there is a long tradition of paternalism, even if patients’ rights have been considerably strengthened in recent decades in psychiatry [34]. Psychiatry is a specific medical discipline that is distinguished both by the centrality of human reflection and by the possibility of treating patients against their will. There is a certain duality between the paternalistic model and the patient autonomy model advocated in the EBENE study [35]. In addition, we had to revise the list of investigating centres after the study had been set up for the first time, following the withdrawal of certain centres. This resulted in a delay in the start of the study and a loss of motivation on the part of some investigators. In some institutions, investigators have not had the time to invest in patient recruitment, instead prioritising their very time-consuming day-to-day clinical work. This last point is linked to the severe budgetary pressures and shortage of medical and paramedical staff in France. This has undoubtedly contributed to the EBENE study taking a back seat in some establishments. All these elements probably contributed to the lack of patient participation in the study. This is especially true since part of the study took place during the COVID period, and it was necessary to protect patients. The pandemic had a strong impact on the functioning of psychiatric and mental health services, causing work-related suffering and affecting mental health services [36]. In this context, research activities were put on hold in many institutions in Europe so that teams could focus on the most urgent care [37]. The COVID-19 pandemic undoubtedly dealt a fatal blow to the already fragile momentum of this study. During this period, professionals prioritised patient protection by limiting contact with other people, especially as the oral sphere was a major source of contamination. Because of the difficulty of including and following up on subjects with schizophrenia, we could not control for biases such as group effect or attrition bias when measuring effectiveness, and we, therefore, chose to present only a descriptive analysis.

Second, a cluster design was chosen to minimize contamination between the intervention and control groups (TEPOH vs. standard care), but structural factors such as the methods used for managing and supporting patients vary from one institution to another. Furthermore, as randomisation took place before patients were included (cluster studies), it is possible that the hospitals or doctors in the control group were less involved in the study, leading to bias and imbalance between the groups. Our previous feasibility study did not assess this point seeing as it was a monocentric study. The complexity of this design and the lack of internal resources might explain the difficulties we encountered in enrolment and follow-up. A weakness in the appropriation of techniques and intervention tools used in the programme may also be partly to blame. Particularly in this study population, we can assume that there is a relationship between patient participation and the extent to which the healthcare team helps participants navigate this research protocol [34].

Third, the choice of our primary endpoint was based on the impact of periodontal disease on oral health. It is recognised that periodontal disease is one of the predictors of oral health (medical and dental status, behaviour, and socioeconomic status) [38]. Although CPI is ideal for long-term follow-up of patients with periodontal disease [39] and is successfully used in regular dental practice [40], this examination can be time-consuming and sometimes painful when the probe is inserted into the periodontal pockets. This may have contributed to the difficulty of including patients with schizophrenia and following up with them over time. Moreover, in the case of EBENE, “hard” assessment tools such as the CPI or the DMFT are open to criticism because they are either too sensitive or not sensitive enough. The intervention did not provide a superior CPI compared to the control. This finding must be considered with caution, given the small size of the population and limitations of the study. More objectively, although our protocol presented solid methodological guarantees, we encountered a feasibility issue that led to a lack of power in our statistical analysis. We have shown in other studies the need to use adapted tools to measure subjectivity in these populations with mental disorders [41]. In the present study, an adapted subjective measurement scale would have made it possible to account for the impact of this intervention in a more detailed way.

## 5. Conclusions

Our research method did not allow us to draw conclusions about the effectiveness of our oral health education programme. However, we found that implementing the programme within a facility supports health prevention and promotion practices, as well as other projects implemented in the facilities. In our case, an adapted subjective measurement scale, rather than the hard criteria we selected, might have enabled us to evaluate in more detail the impact of the intervention. Nevertheless, the long-term follow-up of patients suffering from schizophrenia remains an issue, as is the evaluation of health education and promotion programmes in patients with severe cognitive disorders. Despite some limitations, this study shows the importance of continuing the evaluation of oral health educational interventions with appropriate criteria.

## Figures and Tables

**Figure 1 healthcare-11-01947-f001:**
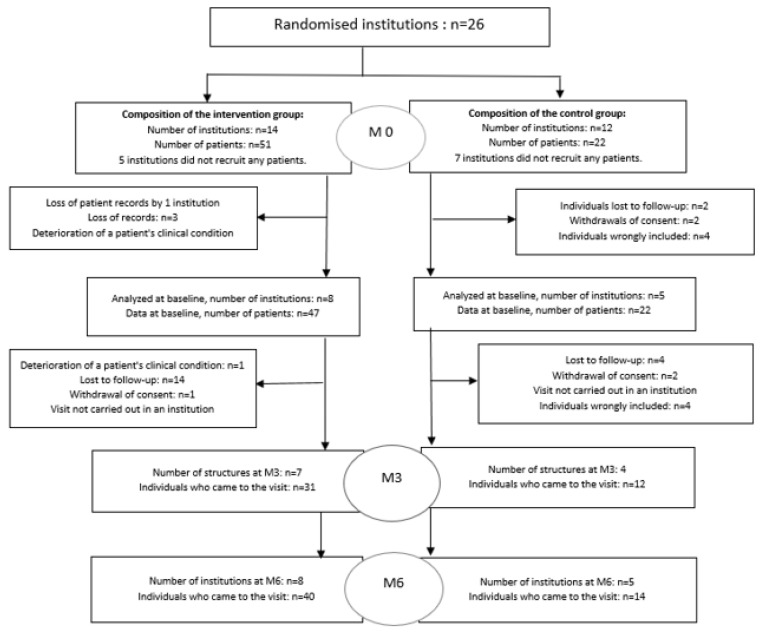
Flow chart of randomised population.

**Table 1 healthcare-11-01947-t001:** Inclusion and exclusion criteria.

Inclusion Criteria	Exclusion Criteria
Persons who have provided consent	Persons not covered by national health insurance
Persons of either sex over 18 years of age	Persons not stabilized from a psychiatric viewpoint or persons in an acute psychiatric episode
Persons with diagnosis of schizophrenia as defined in the Diagnostic and Statistical Manual of Mental Disorders-Fifth edition (DSM-5)	Pregnant or breast-feeding women Edentulous persons Persons hospitalized under stress
Receiving care in hospital (in- or outpatient)	Cannot understand or have a poor understanding of French Patients with risk of infective endocarditis ^a^ or major risk of superinfection People undergoing chemotherapy

^a^: Persons with prosthetic valve, cyanotic congenital heart disease, history of infectious endocarditis.

**Table 2 healthcare-11-01947-t002:** Scale used in the study.

Scale	Expected Findings Out
Global Oral Health Assessment Index (GOHAI)	To detect changes in oral health related quality of life
Schizophrenia Quality of Life Scale (S-QoL)	To detect changes that patient experience in quality of life of PWS
Beck Depression Inventory	To screen for and measure the severity of depressive symptoms. Depressive symptoms are strongly associated with poor oral health

**Table 3 healthcare-11-01947-t003:** Description of study population.

	Overall	Intervention Group	Control Group
Characteristics	*n* = 69	*n* = 47	*n* = 22
Age, years, mean ± sd	44.4 ± 11.6	44.3 ± 11.3	44.7 ± 12.5
Female, *n* (%)	25 (36.2)	17 (36.2)	8 (36.4)
Body mass index, mean ± sd	30.0 ± 6.3	29.3 ± 5.6	31.5 ± 7.6
Lifestyle, *n* (%)			
*In couple or cohabitation*	19 (27.5)	17 (36.2)	2 (9.1)
*Alone*	32 (46.4)	23 (48.9)	9 (40.9)
*Institution*	18 (26.1)	7 (14.9)	11 (50.0)
Level of study ^a^, *n* (%)			
*Primary level*	14 (20.6)	8 (17.4)	6 (27.3)
*Secondary level and above*	54 (79.4)	38(82.6)	16 (72.7)
Time to care ^b^, *n* (%)			
*Less than 5 years*	18 (26.9)	13 (27.7)	5 (25.0)
*Between 5 and 15 years old*	18 (26.9)	9 (19.1)	9 (45.0)
*More than 15 years*	31 (46.2)	25 (53.2)	6 (30.0)
CPI score, mean ± sd	2.0 ± 1.2	2.3 ± 1.2	1.6 ± 1.1
CPI score *≥ 3*, *n* (%)	22 (31.9)	20 (42.5)	2 (9.1)
CPITN, *n* (%)			
*TNO*	10 (14.5)	5 (10.6)	5 (22.7)
*TN1*	8 (11.6)	5 (10.6)	3 (13.6)
*TN2*	42 (60.9)	29 (61.7)	13 (59.1)
*TN3*	9 (13.0)	8 (17.0)	1 (4.6)
DMFT score, mean ± sd	18.1 ± 8.2	17.6 ± 7.3	19.1 ± 9.9
*D*	2.9 ± 3.8	3.4 ± 4.1	1.9 ± 2.9
*M*	8.8 ± 7.1	7.7 ± 5.7	11.1 ± 9.3
*FT*	6.1 ± 5.1	6.2 ± 4.3	5.9 ± 6.5
*OHIS ^d^ Score*, mean ± sd	2.2 ± 1.2	2.2 ± 1.1	2.3 ± 1.3
SQOL ^c^ Score, mean ± sd	59.5 ± 11.1	60.5 ± 9.8	57.2 ± 13.6
*GOHAI ^e^ Score*, mean ± sd	47.3 ± 7.5	47.1 ± 7.3	47.8 ± 8.1
*Beck ^f^ Score*, mean ± sd	8.6 ± 7.2	8.6 ± 7.1	8.4 ± 7.7
Frequency of visits to the dentist, *n* (%)			
*More than once a year*	13 (18.8)	7 (14.9)	6 (27.3)
*occasional*	37 (53.6)	24 (51.1)	13 (59.1)
*Never*	19 (27.5)	16 (34.0)	3 (13.6)
Frequency of brushing teeth, *n*(%)			
*Every day*	48 (69.6)	31 (66.0)	17 (77.3)
*Less than once a day*	13 (18.8)	11 (23.4)	2 (9.1)
*Never*	8 (11.6)	5 (10.6)	3 (13.6)
Consumption of soft drinks			
*Every day*	17 (24.6)	10 (21.3)	7 (31.8)
*occasional*	37 (53.7)	25 (53.2)	12 (54.6)
*Never*	15 (21.7)	12 (25.5)	3 (13.6)
Alcohol consumption ^g^, *n* (%)			
*Every day*	2 (3.0)	0	2 (9.1)
*occasional*	17 (25.8)	13 (29.5)	4 (18.2)
*Never*	47 (71.2)	31 (70.5)	16 (72.7)
Tobacco use, *n* (%)	37 (53.6)	27 (57.4)	10 (45.4)
Recreational drug use, *n* (%)	2 (2.9)	2(4.3)	0
Snacking between meals, *n* (%)			
*Every day*	20 (29.0)	12 (25.5)	8 (36.4)
*occasional*	34 (49.3)	24 (51.1)	10 (45.5)
*Never*	15 (21.7)	11 (23.4)	4 (18.2)

Abbreviations: CPI = Community Periodontal Index, CPITN = Community Periodontal Index of Treatment Needs, DMFT = decayed, missing, and filled teeth, GOHAI = Geriatric Oral Health Assessment Index, OHI-S = Simplified Oral Hygiene Index, SD = standard deviation, SQOL = Subjective Quality Of Life; ^a^ Missing data: *n* = 1 in the intervention group; ^b^ Missing data: *n* = 2 in the control group; ^c^ Missing data: *n* = 3 in the control group; ^d^ Missing data: *n* = 6 in the intervention group and *n* = in the control group; ^e^ Missing data: *n* = 1 in the intervention group and *n* = 1 in the control group; ^f^ Missing data: *n* = 2 in the intervention group and *n* = 1 in the control group; ^g^ Missing data: *n* = 3 in the intervention group.

**Table 4 healthcare-11-01947-t004:** Description of the assessment criteria.

	Intervention Group	Control Group
**CPI index ^a^, *n*(%)**		
Month 3,	*n* = 31	*n* = 12
<3	20 (64.5)	9 (75.0)
≥3	11 (35.5)	3 (25.0)
Month 6,	*n* = 40	*n* = 14
<3	32 (80.0)	12 (85.7)
≥3	8 (20.0)	2 (14.3)
**DMFT score ^a^,**		
Month 3,	*n* = 31	*n* = 12
Mean ± SD	16.8 ± 7.2	17.7 ± 8.9
Change from baseline	0.1 ± 1.5	0.4 ± 0.9
Month 6,	*n* = 40	*n* = 14
Mean ± SD	17.2 ± 7.1	15.4 ± 8.9
Change from baseline	0.1 ± 2.4	0.1 ± 0.7
**OHIS score ^a^,**		
Month 3,	*n* = 31	*n* = 12
Mean ± SD	1.5 ± 1.2	1.9 ± 0.9
Change from baseline	−0.5 ± 0.7	−0.2 ± 0.9
Month 6,	*n* = 40	*n* = 14
Mean ± SD	1.1 ± 1.0	1.9 ± 1.2
Change from baseline	−1.0 ± 1	−0.0 ± 1.1
**SQOL score ^a^,**		
Month 3,	*n* = 21	*n* = 7
Mean ± SD	64.7 ± 9.9	52.0 ± 15.3
Change from baseline	2.8 ± 9.3	−1.4 ± 6.8
Month 6,	*n* = 31	*n* = 10
Mean ± SD	64.0 ± 9.9	60.5 ± 15.5
Change from baseline ^b^	4.1 ± 11.6	4.7 ± 19.7
**GOHAI score ^a^,**		
Month 3,	*n* = 31	*n* = 12
Mean ± SD	48.4 ± 6.7	48.3 ± 5.5
Change from baseline ^c^	0.2 ± 4.9	−1.5 ± 6.4
Month 6,	*n* = 39	*n* = 14
Mean ± SD	48.1 ± 6.2	49 ± 8.1
Change from baseline ^c^	0.3 ± 5.7	0.7 ± 6.7
**Beck score ^a^,**		
Month 3,	*n* = 30	*n* = 8
Mean ± SD	5.9 ± 6.3	6.4 ± 6.5
Change from baseline ^c^	−0.7 ± 2.8	−2.5 ± 4.6
Month 6,	*n* = 37	*n* = 11
Mean ± SD	7.4 ± 6.9	4.8 ± 5.0
Change from baseline ^c^	−1.2 ± 3.9	−1.8 ± 2.9

Abbreviations: CPI = Community Periodontal Index, DMFT = decayed, missing, and filled teeth, GOHAI = Geriatric Oral Health Assessment Index, OHI-S = Simplified Oral Hygiene Index, SD = standard deviation, SQOL = Subjective Quality of Life; ^a^ *n* represents the number of patients with observed data at a specific timepoint for each outcome. ^b^ Missing data: *n* = 2 in the intervention group; ^c^ Missing data: *n* = 1 in the intervention group.

## Data Availability

The data that support the findings of this study are available from the corresponding author upon reasonable request.

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
