# Peer review of "Effectiveness of a Therapeutic Educational Oral Health Program for Persons with Schizophrenia: A Cluster Randomized Controlled Trial and Qualitative Approach"

_healthcare, 2023, doi:10.3390/healthcare11131947_

Round 1
Reviewer 1 Report
Abstract
- Please expand abbreviations if not used before (TEPOH)
- Conclusion needs to rewrtitten: “ we felt it was important” even if the results were not significant, the conclusion should be written accordingly.
Any registration details of the RCT?
The paper is well written and clear. I do not have any suggestions or modifications.

The paper is well written and clear. English editing is required.
Author Response
Comments and Suggestions for Authors
Abstract
1/Please expand abbreviations if not used before (TEPOH).
Response
This correction was made.
2/ Conclusion needs to rewritten: “we felt it was important” even if the results were not significant, the conclusion should be written accordingly.
Response:
We have rewritten the conclusion as requested.
3/Any registration details of the RCT?
Response
This correction has been made in the abstract and in the Manuscript p2 line 91.
4/The paper is well written and clear. I do not have any suggestions or modifications.
Response:
Thank you
Reviewer 2 Report
Thank you so much for allowing me to review this article. There are some comments I would like you to address.
Abstract:
I would recommend you to change the abstract, considering the following aspects.
In my opinion, the results section should express the numerical data obtained that are reflected in the conclusion. Furthermore, the method to be used to evaluate the effectiveness of the programme is not explained. I would also recommend you to change the conclusions of the study and remove the part related to the decision to publish the results even if they are not in line with expectations.
Introduction
Line 42, 66, 69 and 86: you should correct the expression “people with PWS”
In my opinion, it is not necessary to explain each of the stages of the EBENE study in the introduction section. It would be more informative for readers if you would explain what the Therapeutic Educational Oral Health Program is and why such interventions are required and appropriate for PWS.
Please, describe the objective of the study at the end of the introduction section to make it clearer.
Materials and Methods
A section describing the study population would be appreciated under the heading of study settings. It is not very clear from the text what the educational intervention and standard care consists of and how the effectiveness of the intervention will be assessed. Please elaborate on these issues.
Another issue that, in my opinion, should be better explained to readers, is why they chose the CPI or DMFT or OHI-S index to assess effectiveness. In my opinion, it is necessary to explain what these indices reflect and evaluate.
Results:
After reading this section, I have some doubts about the comparison of the groups that I would be grateful if you could explain to me. Why is the size of the two groups not similar? What happened to the subjects who did not attend the 3-or 6 month appointment?
Discussion:
In my opinion, the most interesting part of the evaluation of the study is related to the qualitative method and is not reflected in the discussion section.
References:
Please check bibliographic references 8 and 9 because they are the same.
Author Response
Thank you so much for allowing me to review this article. There are some comments I would like you to address.
Response:
Thank you
Abstract:
1/I would recommend you to change the abstract, considering the following aspects.
In my opinion, the results section should express the numerical data obtained that are reflected in the conclusion. Furthermore, the method to be used to evaluate the effectiveness of the programme is not explained. I would also recommend you to change the conclusions of the study and remove the part related to the decision to publish the results even if they are not in line with expectations.
Response
We thank the reviewer for this suggestion. The changes have been made as requested.
Introduction
2/Line 42, 66, 69 and 86: you should correct the expression “people with PWS”.
Response
This correction has been made.
3/In my opinion, it is not necessary to explain each of the stages of the EBENE study in the introduction section.
Response:
As requested, we deleted lines 95 to 98.
4/ It would be more informative for readers if you would explain what the Therapeutic Educational Oral Health Program is and why such interventions are required and appropriate for PWS.
Response:
We introduced new sentence page 2 lines 84 to 85 to explain this point and added a new reference.
- Gremyr A, Andersson Gäre B, Thor J, Elwyn G, Batalden P, Andersson AC. The role of co-production in Learning Health Systems. Int J Qual Health Care. 2021 Nov 29;33(Supplement_2):ii26-ii32. doi: 10.1093/intqhc/mzab072. PMID: 34849971; PMCID: PMC8849120.
5/Please, describe the objective of the study at the end of the introduction section to make it clearer.
Response:
We introduced a new sentence at the end of the “Introduction” section.
Materials and Methods
6/A section describing the study population would be appreciated under the heading of study settings.
Response:
We have introduced these new sentences on page 6, lines 203 to 205.
“Depending on the protocol, the number of PWS to be included with a type-I error of 5 % and type-II error of 20 % were to 202 patients to highlight a statistically significant difference between the two groups [16]. “
7/It is not very clear from the text what the educational intervention and standard care consists of and how the effectiveness of the intervention will be assessed. Please elaborate on these issues.
Response:
The educational intervention is a process of reinforcing the patient's ability to manage their oral hygiene. The educational intervention is an integral part of therapeutic management. It is complementary to and inseparable from treatment and care. The effectiveness of the educational intervention is based on various criteria described in chapter 2.5. We have included these explanations page 4 lines 172 to 176.
8/Another issue that, in my opinion, should be better explained to readers, is why they chose the CPI or DMFT or OHI-S index to assess effectiveness. In my opinion, it is necessary to explain what these indices reflect and evaluate.
Response:
Thank you for this suggestion.
The CPI index quantifies the degree to which the periodontium is affected by periodontal disease. It is the benchmark indicator of periodontal health and is closely linked to patients' level of hygiene.
The DMFT, this index gives the sum of an individual's decayed, missing, and filled permanent teeth and provides an indicator of both current and past caries experience.
The OHI-S indices help dentists to determine a patient's level of oral hygiene by scoring debris and calculus accumulation in the mouth.
We precis the points in chapter 2.5. Outcomes.
Results:
9/After reading this section, I have some doubts about the comparison of the groups that I would be grateful if you could explain to me. Why is the size of the two groups not similar?
Response:
The lack of similarity in the number of patients and their characteristics between the two arms of the study can be attributed for the most part to the following factors:
- Cluster randomization, which takes place prior to subject inclusion. Cluster characteristics are then evenly distributed between groups, but this is not necessarily the case for the characteristics of the individuals composing each cluster.
- Lack of blinding, which was not possible due to the nature of the study intervention. Since randomization takes place prior to patient inclusion (cluster studies), it's common for hospitals or doctors who are assigned an intervention that doesn't suit them (e.g. if they're randomized to a control group) to ultimately refuse to take part in the study, obviously inducing bias and exacerbating the risk of imbalance between groups.
The resulting imbalance is usually taken into account in the analyses using appropriate statistical methods (generalized logistic regression models).
We explained these points in the discussion section page 14 lines 457 to 459.
10/ What happened to the subjects who did not attend the 3-or 6-month appointment?
Response:
The subjects who did not attend the 3-or 6-month appointment were considered as lost to follow-up. Some of this loss could be explained by the lack of professional involvement during follow-up. However, another part is due to fact that subjects with schizophrenia often do not attend study visits. This effect was exacerbated by the Covid context.
Because of the difficulty of including and following-up subjects with schizophrenia, we couldn’t control for biases such as group effect or attrition bias when measuring effectiveness, and we therefore chose to present only a descriptive analysis.
We explained these points in the discussion section page 14 lines 451 to 453.
Discussion:
11/In my opinion, the most interesting part of the evaluation of the study is related to the qualitative method and is not reflected in the discussion section.
Response:
We agree. We revised the discussion. Please see Page 13, lines 421 to 437.
12/References:
Please check bibliographic references 8 and 9 because they are the same.
Response:
Thank you for informing us of this error. Reference 9 has been modified and listed in the references section.
9-Suvisaari J, Partti K, Perälä J, Viertiö S, Saarni SE, Lönnqvist J, Saarni SI, Härkänen T. Mortality and its determinants in people with psychotic disorder. Psychosom Med. 2013 Jan;75(1):60-7. doi: 10.1097/PSY.0b013e31827ad512. Epub 2012 Dec 20. PMID: 23257931.
Reviewer 3 Report
Dear authors,
Thank you for your study.
I would suggest the following for presenting better your work:
Abstract
-background: Too long phrases. Please redefine the introduction sentence. Explain better the aim, use smaller sentences.
-methods: explain better the methodology. it is not obvious that the survey is based on a previous one/questionnaire and what exactly you did this time, for how long etc. you do not mention you interviewed patients. You only mention this in the results part.
results: delete "however" at the beginning of the sentence. Please rearrange, explain better the groups of the study
lines 68-70: it doesn't make sense
lines 68-81 : remove this to methodology part.. rearrange then your aim and the research questions.
Put a background section just after introduction discussing educational/coaching interventions and studies for minorities' oral health status, methodologies used and possible outcomes. What your study meant to add to the field (despite the fact that you have stopped it) according to these interventions. what did you do differently..
Methodology
92-95: smaller sentences. we miss the meaning of what you want to say
99-100: delete the journal name... Just put the reference
112: you mentioned inclusion and exclusion criteria but those are not well defined. later in the paper you mentioned that you used strict criteria. Please describe better in the methodology part
Also design an new table with all the scales you used in the study and expected findings out of each one. Put also in this table all relevant references.
table 1: there are words missing, poor quality of figure
table 2: fix spaces, check numbers, poor quality of figure
Limitations of the present study: to be discussed further, more analytically
Thank you
Minor spelling check needed
Author Response
Dear authors,
Thank you for your study.
I would suggest the following for presenting better your work:
Response
Thank you
Abstract
1/background: Too long phrases. Please redefine the introduction sentence. Explain better the aim, use smaller sentences.
Response:
We have made change as requested.
2/methods: explain better the methodology. it is not obvious that the survey is based on a previous one/questionnaire and what exactly you did this time, for how long etc. you do not mention you interviewed patients. You only mention this in the results part.
Response:
Thank you for your suggestions. We have made changes as requested.
3/results: delete "however" at the beginning of the sentence.
Response:
The changes have been made as requested.
4/Please rearrange, explain better the groups of the study.
Response:
Many changes have been made.
5/lines 68-70: it doesn't make sense.
Response:
This sentence has been revised.
6/lines 68-81: remove this to methodology part. rearrange then your aim and the research questions.
Response:
Many changes have been made as requested.
7/Put a background section just after introduction discussing educational/coaching interventions and studies for minorities' oral health status, methodologies used and possible outcomes. What your study meant to add to the field (despite the fact that you have stopped it) according to these interventions. what did you do differently.
Response:
We have added the following sentence: “The oral health of people with schizophrenia (PWS) is very poor. Carers need oral health promotion programmes with a high level of evidence to help PWS “. Please see Page 1 lines 16.
Methodology
8/92-95: smaller sentences. we miss the meaning of what you want to say.
Response
We have made change as requested.
9/99-100: delete the journal name... Just put the reference.
Response:
Thank you for this suggestion.
10/112: you mentioned inclusion and exclusion criteria but those are not well defined. later in the paper you mentioned that you used strict criteria. Please describe better in the methodology part.
Response:
We have revised the Table 1 and the chapter “2.3. Recruitment of Centers and Participants” Page 4 lines 164 to 166.
We have added a new reference which we have listed in the references section.
[19]- American Psychiatric Association: desk reference to the diagnostic criteria from. DSM-5; 2013.
11/ Also design a new table with all the scales you used in the study and expected findings out of each one.
Response
As requested, we have added a new table entitled Table 2 on page 5.
12/ Put also in this table all relevant references.
table 1: there are words missing, poor quality of figure
Response:
We have revised the Table 1. Thank you for pointing out this error.
13/table 2: fix spaces, check numbers, poor quality of figure
Response:
When submitted, the source document is distorted by the system. The source document provided separately is of better quality and we hope that it will meet your expectations. We have corrected the errors pointed out and Table 2 is now Table 3.
14/Limitations of the present study: to be discussed further, more analytically.
Response:
As requested, major changes have been made to the discussion section.
Thank you
Response:
We would like to thank the reviewer for his valuable comments, which helped us to improve our manuscript.
Round 2
Reviewer 2 Report
Dear authors,
I thank you for the changes made to the manuscript, but I still have doubts about the method for assessing the effectiveness of the intervention and the validity of the results. My main concerns are due to the loss of subjects during follow-up and at the time of recruiting the population. Why did 12 centres not want to participate? I think it is important that the reasons are included in the discussion. Another question I have is what happened to the patients who did not attend the 3-month visit? Is it likely that the COVID-19 pandemic played a role?
Author Response
Reviewer 2:
1/I thank you for the changes made to the manuscript, but I still have doubts about the method for assessing the effectiveness of the intervention and the validity of the results. My main concerns are due to the loss of subjects during follow-up and at the time of recruiting the population. Why did 12 centres not want to participate? I think it is important that the reasons are included in the discussion.
Response:
We have added a chater on Page 14, lines 445 to 452, to clarify this point.
“In addition, we had to revise the list of investigating centres after the study had been set up for the first time, following the withdrawal of certain centres. This resulted in a delay in the start of the study and a loss of motivation on the part of some investigators. In some insti-tutions, investigators have not had the time to invest in patient recruitment, prioritising their very time-consuming day-to-day clinical work. This last point is linked to the severe budgetary pressures and shortage of medical and paramedical staff in France. This has undoubtedly contributed to the EBENE study taking a back seat in some establishments.”
2/Another question I have is what happened to the patients who did not attend the 3-month visit? Is
Response:
We considered them as "lost to follow-up" patients for the study. Of course, this did not affect the follow-up of their psychiatric pathology.
3/ it likely that the COVID-19 pandemic played a role?
We agree: The COVID-19 pandemic undoubtedly dealt a fatal blow to the already fragile momentum of this study. During this period, professionals prioritised patient protection by limiting contact with other people, especially as the oral sphere was a major source of contamination. We have added this sentence Page 14 lines 458 to 461.
We would like to thank you for your constructive comments, which enabled us to improve our manuscript.
Reviewer 3 Report
Dear authors
The article is much improved. Well done
Minor spelling mistakes. Just go through the whole text for final revision.
Thank you
Minor spelling corrections needed
Author Response
Reviewer 3:
The article is much improved. Well done
Response:
Thank you.
Minor spelling mistakes. Just go through the whole text for final revision.
Response:
A proofreading was carried out.
Thank you
Response:
We would like to thank you for your constructive comments, which enabled us to improve our manuscript.